# NHC-Ni catalyzed enantioselective synthesis of 1,4-dienes by cross-hydroalkenylation of cyclic 1,3-dienes and heterosubstituted terminal olefins

Yang Chen [1], Liang Dang [1✉] & Chun-Yu Ho [1✉]

Enantioenriched 1,4-dienes are versatile building blocks in asymmetric synthesis, therefore their efficient synthesis directly from chemical feedstock is highly sought after. Here, we show an enantioselective cross-hydroalkenylation of cyclic 1,3-diene and hetero-substituted terminal olefin by using a chiral [NHC-Ni(allyl)]BAr$^F$ catalyst. Using a structurally flexible chiral $C_2$ NHC-Ni design is key to access a broad scope of chiral 1,4-diene **3** or **3'** with high enantioselectivity. This study also offers insights on how to regulate chiral $C_2$ NHC-Ni(II) 1,3-allylic shift on cyclic diene **1** and to build sterically more hindered endocyclic chiral allylic structures on demand.

¹ Shenzhen Grubbs Institute, Guangdong Provincial Key Laboratory of Catalysis, Department of Chemistry, Southern University of Science and Technology (SUSTech), Shenzhen 518055, China. ✉email: dangl3@sustech.edu.cn; jasonhcy@sustech.edu.cn

Other than asymmetric allylic substitutions and Diels-Alder reactions[1–3], cyclic 1,3-diene chiral insertion into a transition metal catalyst represents one of the most important ways to build endocyclic chiral allylic structures for drugs synthesis[4]. Significant advances along this line have been made by using M-boryl species, in which notable chiral ligand designs fit at low-substitution degree have been established[5–10] (Fig. 1a). Yet, chiral M-Bpin insertion is less employed or often failed in higher substituted cyclic 1,3-diene[8] due to the significantly different demand in steric, 1,3-allylic shift control, and insertion reactivity. Sterically less demanding chiral M–H insertion approaches can be a good solution, indeed high ee (ee = enantiomeric excess) has been achieved by using chiral phosphorus ligand and polar hydride sources as pairs for a redox-active cycle occasionally[11–14] (Fig. 1b). Non-redox-active catalytic cycle and/or those use simple alkenes directly for green advances are rare[15–23] (Fig. 1c). Very high ee and yield were observed in vinylcyclohexenes or 1-substituted cyclic 1,3-diene with ethylene by using well-designed chiral phosphorus ligand[24,25]. Yet, many easily accessible and commonly observed 2- or higher substituted cyclic 1,3-diene, as well as structurally diverse terminal olefin, are not ideal substrates in those systems. Complications related to undesired steric competitions and terminal olefin consumptions by both facile isomerization and oligomerization are hard to solve[26]. Overall, the above factors have made the former designs inapplicable for some new challenges, like catalytic synthesis of highly substituted dienes with endocyclic chiral allylic structures at high atom economy.

Recently, we have discovered a cross-hydroalkenylation of cyclic 1,3-diene 1 with heterosubstituted terminal olefin 2 by using achiral IPr-Ni(II) as a catalyst[26–30] (IPr = 1,3-bis(2,6-diisopropylphenyl)−1,3-dihydro-2H-imidazol-2-ylidene). It prompted us to explore the potentials of using chiral NHC for solving the above problems (NHC = N-heterocyclic carbene). Herein, we develop a [chiral C$_2$ NHC–Ni(allyl)]BArF catalyst (BArF = [3,5-bis(trifluoromethyl)phenyl]borate) for enantioselective cross-hydroalkenylation of 1 with 2 (Fig. 1d). It provides an efficient access to synthetically valuable skipped 1,4-dienes[14,24,31,32] with a chiral allylic center and endo-olefin, at high ee and high regiocontrol. Using chiral C$_2$ NHC with low substitution on N-aryls is a key for an effective chiral induction, a 1,3-shift control and a better scope of cyclic 1,3-diene[33–35]. Compared with synthetic alternatives that are based on alkenyl metal, this work overcomes challenges related to synthesis and reactivity of different leaving groups[36–39].

## Results

**Chiral catalyst development and optimization.** Our study commenced with a screening of a set of chiral [L-Ni(allyl)]BArF catalysts (Fig. 2, 5 mol%) by using 1a and 2a (Table 1, entries 1–8, monosubstituted dienes, entries 1–3. $R^2$ = OTES (TESO = triethylsiloxyl), which is easily accessible from α,β-unsaturated ketones, yet a rarely explored substrate). High yield and regioselectivity of the desired chiral 1,4-diene 3aa were observed by using L1 (entry 1). It featured a C$_1$ chiral environment, and was identified as a benchmark chiral NHC for enantioselective tail-to-tail cross-hydroalkenylation of styrene with terminal olefin before[29,30]. Yet, only moderate ee was obtained. Using bulkier C$_2$ chiral NHC did not help, a drop in both ee and desired reactivity were noted (L2, entry 2). It has been suggested that there might be undesired steric interactions among 2-substituted cyclic 1,3-diene and the bulky rigid NHCs employed above. So, a less bulky and more flexible C$_2$ chiral NHC was tested next, in hope the substituent effects on the 2-substituted cyclic 1,3-diene could work with NHC together (L3, entry 3). To our delight, it provided a good balance of ee and yield without losing the high-insertion regioselectivity or causing a faster and undesired consumption of 2a. Further study showed that the size and the place of R on cyclic 1,3-diene are both very critical for achieving a good balance of desired reactivity and selectivity by using chiral NHC (entries 3–5 and 6–8). In general, higher ee could be obtained when 1 with bulky $R^2$ rather than $R^1$ (entry 3 vs 6, and 3 vs 4). This set of observation suggested that could be a result of an ineffective steric effect cooperation in Ni–H insertion step, and/or a facile NHC–Ni (II) 1,3-allylic shift when $R^2$ is smaller than OTES. Yet and notably, by comparison with the optical rotation values of related structures, an efficient 1,3-allylic shift was occurred after insertion on 1a (see Supplementary Table 4 and Supplementary Fig. 8).

**Scope of the enantioselective cross-hydroalkenylation and 1,4-diene.** Above findings prompted us to elucidate other factors on cyclic 1,3-diene 1 that might favor high ee and to give greater structural diversity of 1,4-diene 3 by using chiral NHC (Table 1, entries 9–12, 1,2-disubstituted dienes). Several notable features and tactics were revealed by this study. First, in sharp contrast to the above and most of the related literature, this study showed that using large OSiR$_3$ at $R^2$ is not mandatory for achieving high ee and 3:4 ratio at here. High selectivity also can be obtained by using 1,2-disubstituted cyclic 1,3-diene 1 (entry 9–12). The higher ee observed in 1 g vs 1a was attributed mainly to a more difficult

**Fig. 1 General chiral insertions tactics for catalytic chiral allylic center preparations using 1,3-dienes. a** Boration by chiral L-M-Bpin. **b** Hydrosilylation, -thiolation, and -boration. **c** Hydro-vinylation by (BDPP)CoCl$_2$ (1,4-insertion, exocyclic). **d** This work: hydroalkenylation by [chiral NHC–Ni (allyl)]BArF.

**Fig. 2 NHC structures employed in this work. L1** chiral C$_1$ NHC with unsymmetric N-aryl substituents. **L2** chiral C$_2$ NHC with highly substituted N-aryl groups. **L3** chiral C$_2$ NHC with 2-substituted N-aryl groups.

**Table 1 Screening of NHC for enantioselective cross-hydroalkenylation[a].**

Reaction conditions: $1x$ + $2a$ ($1x$:$2a$ = 1:3), 5 mol% (0.05 mmol) [L-Ni(allyl)]BAr$^F$, 1-octene (20 mol%), toluene (4 mL), 6 h, r.t. → $3xa$ + $4xa$ (with OBn substituents)

| Entry | L | 1 | R$^1$/R$^2$ | 3 | Yield % | 3:4 | % ee 3 |
|---|---|---|---|---|---|---|---|
| 1 | L1 | 1a | H/OTES | 3aa | >95 | 95:5 | 78[c] |
| 2 | L2 | 1a | H/OTES | 3aa | 36 | 95:5 | 54[c] |
| 3 | L3 | 1a | H/OTES | 3aa | >95 | 95:5 | 87[c] |
| 4 | L3 | 1b | H/n-Oct | 3ba | >95 | 95:5 | 6[c] |
| 5 | L3 | 1c | H/OTMS | 3ca | >95 | 80:20 | 75[c] |
| 6 | L3 | 1d | OTES/H | 3da | >95 | 71:29 | 48 |
| 7 | L3 | 1e | OMe/H | 3ea | 84 | 75:25 | 39 |
| 8[b] | L3 | 1f | OTIPS/H | 3fa | 78 | 82:18 | 29 |
| 9 | L3 | 1g | Me/OTES | 3ga | >95 | 68:32 | 92, 98[d] |
| 10 | L3 | 1h | Me/OTMS | 3ha | >95 | 75:25 | 86 |
| 11 | L2 | 1i | OTES/Me | 3ia | >95 | >95:5 | 90 |
| 12 | L3 | 1i | OTES/Me | 3ia | >95 | >95:5 | 96 |

aGeneral reaction condition: cyclic 1,3-diene 1 and heterosubstituted terminal olefin 2 were added to the indicated catalyst and stirred at r.t. for 6 h. Yields and selectivity were determined by 1H NMR, HPLC, and isolation. Product with −ve optical rotation was obtained.
b24 h.
c+ve optical rotation value was obtained (see Fig. 5).
dee of chiral 1,4-diene 4.

1,3-allylic shift and the restricted OTES rotation by the Me at R$^1$ (entry 3 vs 9). More dramatically, ee value was doubled when 1i over 1d was used (i.e., R$^2$ = Me vs H, 96% vs 48% ee). So, the steric effect of R$^1$ and R$^2$ cooperates, where the choice of R$^2$ can be as small as a Me, and R$^1$ can be inadequate in size to favor a high ee insertion by itself alone.

Further screening showed that the above cooperation is invaluable in expanding scope to a broader range of 1,4-diene 3 at high ee. It represents a general high ee method to avoid extensive use of very bulky R$^2$ or R$^1$ on cyclic 1,3-diene (Fig. 3), which is a major obstacle and limitation in many cases before. It accepts a good variety of R$^2$ and functional groups, like higher substituted alkyls, ethers, and Bn without impairing the desired selectivity at high ee and yield (Set 1a, 88–97% ee, 3:4 > 95:5 in most cases). The success of using Br as R$^2$ is notable since it can be a useful handle for cross-coupling and as a masked H by reduction. So, it offers a key for having greater 3 structural varieties at high ee. Heterocyclic substituent and 1,4-disubstitutions also work well with L3-Ni(II) catalyst for desired reactivity and high ee (Fig. 3, Set 2–3, 84–98% ee). At here, two new stereocenters could be obtained in 1,4-substituted diene at trans-selectivity. At this stage, cyclic 1,3-diene substituted with a ketone did not work well.

The fine-tuned chiral L3-Ni(II) reactivity and chiral induction ability on substituted cyclic 1,3-diene came with also a broad scope of 2 (Fig. 3, Sets 4–6). High ee, 3:4 ratio and yield were observed under general condition. That is not straightforward since inefficient insertion of terminal olefin 2 by undesired steric interactions could lead to backward reaction, lower yield, and ee. Competing pathways, like isomerization/oligomerization of 2 and insertion of 2 on hydrometallated cyclic 1,3-diene minor enantiomer, could be more competitive. Yet, linear, branch, cyclic, aliphatic, and aromatic groups on heterosubstituted terminal olefin 2 were found quite effective (88–97% ee, >95% yield, 3:4 > 95:5). Only oversized NBnTs and protic NHTs (Bn = benzyl, Ts = tosyl) on heterosubstituted terminal olefin 2 were found inefficient and incompatible, respectively. Notably, using optimal size of heterosubstituted terminal olefin 2 was found useful in scope expansion of cyclic 1,3-diene and in selectivity improvement. Indeed, higher ee, yield, and 3:4 ratio were achieved by using allylNMeTs or allylOPh over 2a in Set 1b vs Table 1 entry 9 (3:4 = >95:5 vs 68:32, 92–96% ee). Such advance is mainly attributed to a relatively slower undesired 2 consumption, and a higher steric demand in heterosubstituted terminal olefin 2 insertion to the minor enantiomer of the hydrometallated cyclic 1,3-diene. Our design also allowed chemoselective insertion of terminal over internal or gem-substituted allylic ether (Set 5). It was shown that those diallyl ether 2 cycloisomerizations[40] were not as competitive as our desired intermolecular skipped diene 3 formation. Heteroatom effect of 2 is important on 3:4 ratio (Set 6)[26].

Higher substituted cyclic 1,3-diene with R$^5$ or R$^6$ were examined in hope to elucidate how those rarely explored substitution patterns in the literature might affect our chiral NHC system (Set 7). Indeed, their effects on both reactivity and ee are quite strong. Some of those patterns were found devastating to our L3-Ni(II) catalyst chiral induction effects and reactivity when cyclic 1,3-diene substituted with R$^1$ = OTES was used (e.g., Set 7, 4–67% ee). Yet, the cooperation of R$^1$ and R$^2$ was found applicable, and 96% ee was achieved.

Finally, further study on the substitution patterns on cyclic 1,3-diene and properties on heterosubstituted terminal olefin have revealed a fascinating way to control NHC–Ni(II) 1,3-allylic shift on the rings (Set 8). A high selectivity of 3:3′ and vice versa were noted in several cases of 1 with R$^1$ = H, e.g., 5–7 member cyclic dienes (3′ refers to 1,3-shifted product)[28]. Besides steric effect on cyclic 1,3-diene, the results showed that both steric and electronic properties of heterosubstituted terminal olefin are related to

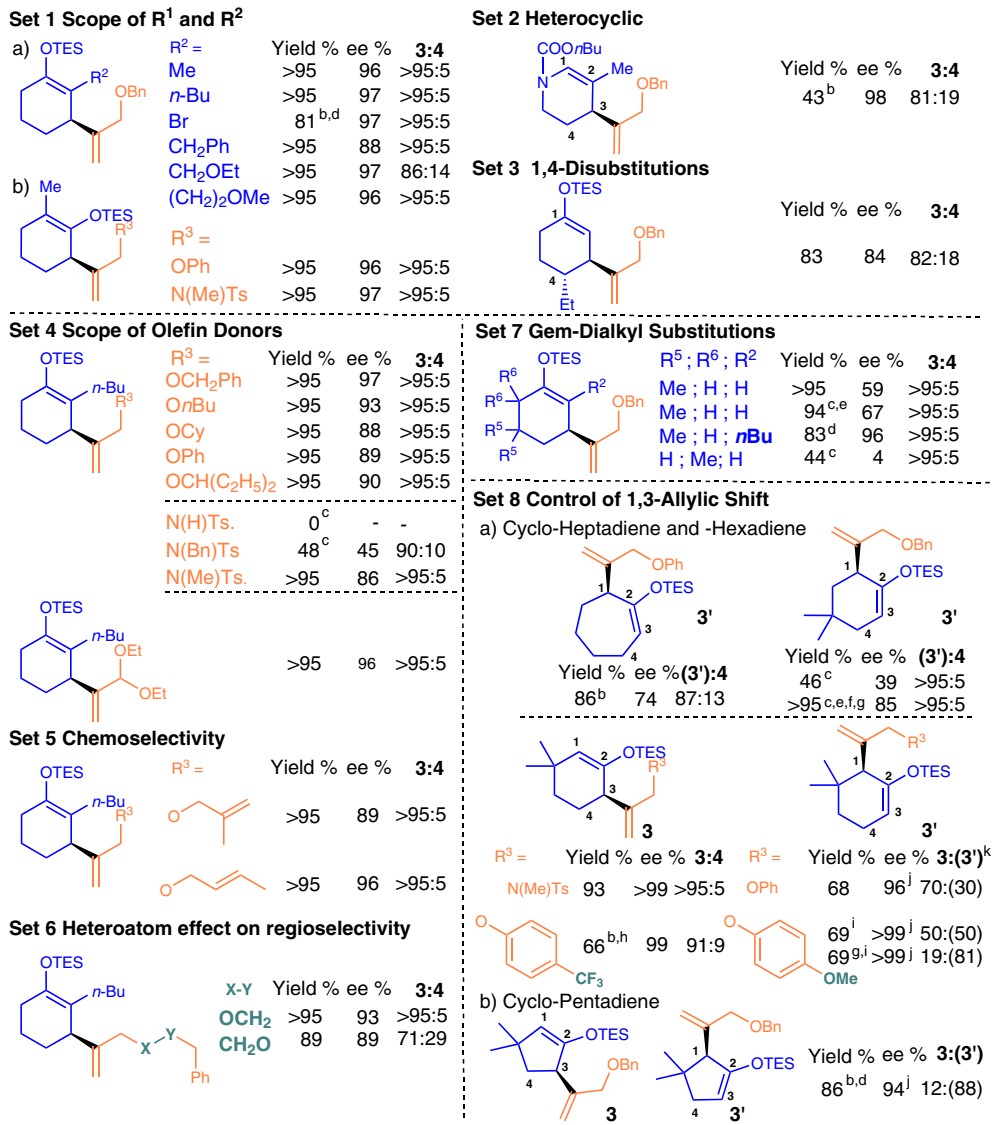

**Fig. 3 Scope of the enantioselective cross-hydroalkenylation[a].** Superscript "a" indicates the standard condition employed in this figure: [**L3**-Ni(allyl)]BArF catalyst (5 mol%, 0.05 mmol), cyclic 1,3-diene **1**: heterosubstituted terminal olefin **2** = 1:3, in 4 mL of toluene at r.t. for 6 h. Yield, chiral 1,4-diene **3:4** or **3:3'** or **3':4** ratio were determined by [1]H NMR spectroscopy, except otherwise indicated. No chiral 1,4-diene **3'** was observed, except in Set 8. Major regioisomer's structure was confirmed by isolation and enantiomeric excesses were determined by HPLC. Superscript "b and c" indicate the amount of catalyst employed, which is 10 mol% and 20 mol%, respectively. Superscript "d" indicates an extra 1 equiv. of heterosubstituted terminal olefin **2** was added after 2 h. Superscript "e" indicates **L1** was used instead of **L3**. Superscript "f" indicates that heterosubstituted terminal olefin **2** was added in three equal portions in 6 h. Superscript "g" indicates taht the reaction was carried out at 0 °C. Superscript "h" indicates the ratio of cyclic 1,3-diene **1** to heterosubstituted terminal olefin **2** here was 1:1.5, and reaction time was 11 h. Superscript "i" indicates that reaction time was 15 h. Superscript "j" indicates the ee value of chiral 1,4-diene **3'**. Superscript "k" indicates that **4** was also observed in those entries and in 27, 17, and 8% yield, respectively.

1,3-allyl shift regulation [as reflected by **3:3'** ratio, $R_3$ = NMeTs, $O(C_6H_4)$p-OMe vs p-CF$_3$] on a given structure of cyclic 1,3-diene and NHC–Ni(II) catalyst. This feature is uniquely useful as the heterosubstituted terminal olefin regioselective insertion involved in forming **3'** is sterically more shielded than in **3** and **4** by simple inspection. Thus, 1,3-allylic shift on the ring with this **L3**-Ni(II) species may not be so highly reversible as typically assumed under this circumstances, and in turn affected the regioselective insertion of heterosubstituted terminal olefin.

## Discussion

At this stage, the keys for the highly asymmetric synthesis of skipped diene **3** with broad scope are attributed mainly to (i)

optimal chiral NHC–Ni(II) insertion reactivity, (ii) an effective 1,3-allylic shift control, and (iii) a highly regioselective insertion reactivity of heterosubstituted terminal olefin by the cooperation of chiral NHC–Ni and substrate, in which all of them were done under mild condition (Fig. 4, Models a–c and d, respectively). Here, the chiral $C_2$ NHC adopts a conformation as shown in Fig. 4, Models a–c[41–43], which is correlated to the absolute configuration of **3** obtained (see Supplementary Fig. 8). Effective steric interaction among cyclic 1,3-diene and Cy on the **L3** N-aryl determines the Model a–c ratio. Due to the reversibility of cyclic 1,3-diene hydrometallation, an ineffective insertion of heterosubstituted terminal olefin to Model a and more effective insertion of heterosubstituted terminal olefin to Model b and c lower the product ee, and vice versa (e.g., NMeTs vs NBnTs).

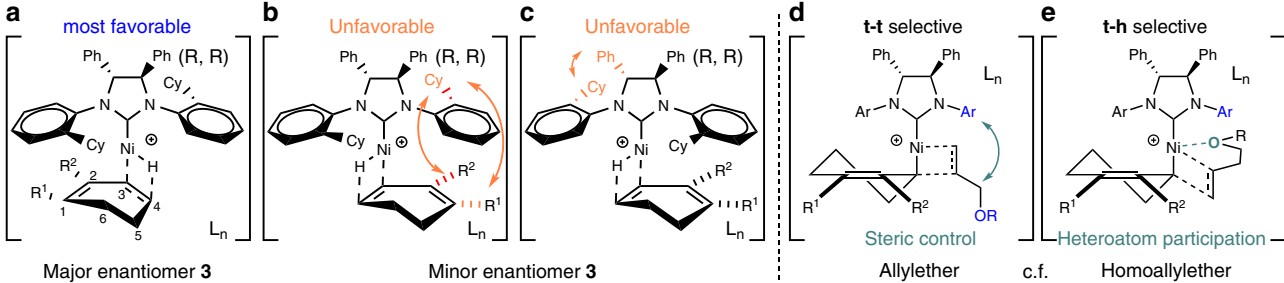

**Fig. 4 Hypothetical Models for chiral insertion and regioselective insertion. a** Chiral NHC–Ni(II)H insertion to diene which give major enantiomer 3. **b**, **c** Unfavorable steric interactions among NHC and diene which disfavored the insertion. **d** Steric control regioselective insertion of terminal olefin. **e** Role of heteroatom in regioselective insertion of homoallylether.

**Fig. 5 Effect of chiral C$_2$ NHC–Ni(II) on 1,3-allylic shift.** 1,3-allylic shift from 3- to 1-position is favored.

**Fig. 6 Chiral α,β-substituted cyclohexanone from 3.** Structures were assigned by comparison with literature. **a** Condition for anti-product preparation. **b** Condition for syn-product preparation.

In addition to the above, the formation of **3′** at high ee may occur when R$^1$ = H. That is attributed to a relatively slower insertion of terminal olefin to Models a–c, and a faster **L3**-Ni(II) 1,3-allylic shift than backward reaction. Indeed, the allylic shift of **L3**-Ni(II) from 3- to 1-position on cyclic 1,3-diene is quite favorable in general, even in the absence of help from R$^5$ and/or R$^6$ (Fig. 5 vs Fig. 3, Set 8). In other words, the higher ee obtained in R$^2$ = OTES (Table 1 entry 3 vs 4, 5) was not a direct result of steric suppressed NHC–Ni(II) 1,3-allylic shift. It should be attributed to the difference in insertion efficiency of heterosubstituted terminal olefin (C–C forming step) before and after the 1,3-shift, as well as the degree of the concerned 1,3-shift equilibrium, likely by creating a different steric insertion demand with the help of the chiral NHC environment. Notably, the high **3′:3** selectivity or vice versa are not controlled by steric effects only. Those are tunable also by the electronic properties of heterosubstituted terminal olefin (Set 8a, R$^3$ = OAryl). This shows that using a more electron-rich alkene is possible to overcome the undesired steric insertion barrier at the more hindered 1-position. And the **L3**-Ni(II) design strongly favors 1- over the 3-position on 2-substituted cyclic 1,3-diene, particularly at lower temperature.

According to Model d, preparation of **4** might be enabled by stronger steric repulsion among substituents on heterosubstituted terminal olefin and R$^2$ on cyclic 1,3-diene 1 vs NHC. Indeed, the high **3:4** ratio was diminished when **1i** was used in place of **1d** (Table 1, entry 12 vs 9, >95:5 vs 68:32). Moreover, the use of homoallylic over allylic ether as heterosubstituted terminal olefin was found useful to assist **4** formation under otherwise the same condition (Fig. 3, Set 6, >95:5 vs 71:29). At this stage, it was attributed mainly to a potential Ni-ether coordination (Model e).

Synthetically, the silyl enol ether on **3** gives us a versatile handle for post modifications (for alkylation example, see ref. [44]; for aldol reaction example, see ref. [45]; for Paterno-Buechi reaction example, see ref. [46]. Extra chiral centers can be acquired expediently at high efficiency by those robust chemistry. For instance, both syn- and anti-products were made selectively by simple deprotection (Fig. 6).

In conclusion, an enantioselective cross-hydroalkenylation of cyclic 1,3-diene 1 and heterosubstituted terminal olefin 2 to give 1,4-diene **3** or **3′** with endocyclic chiral allylic structures was established by manipulating the relative insertion reactivity of 1 and 2 on chiral NHC–Ni(II) catalyst (up to 99% ee, 86–96% ee in general). Here, the design covers a broad range of 1 and shows a

good ability in chiral NHC–Ni(II) 1,3-allylic shift regulation, where both are keys in related asymmetric NHC–Ni catalysis development in general. Selective synthesis of sterically crowded diene **3′** and the electronic effect of heterosubstituted terminal olefin 2 on regioselective insertion are unexpected, which have prompted further study in near future. Overall, this work has shed light on how to use chiral NHC–Ni for chiral allylic centers preparation and for better chiral 1,4-dienes availability from cyclic 1,3-diene 1 and heterosubstituted terminal olefin 2 directly.

## Methods
**General procedure for enantioselective cross-hydroalkenylation**. To a [**L3**-Ni(Allyl)]BAr$^F$ catalyst in toluene (0.05 mmol, 5 mol%, 2 mL) with 20 mol% 1-octene (for generation of NiH or equiv.)[26–30], a toluene solution of cyclic 1,3-diene 1 and heterosubstituted terminal olefin 2 (1:3, 2 mL) was added in one-pot and stirred at r.t. for 6 h. After work up, the 1,4-diene yield, structure and selectivity (**3:4** or **3:3'**) were determined by $^1$H NMR, HPLC and isolation (average of two runs).

## Data availability
All data generated during/or analyzed during the current study are included in this published article and its supplementary files.

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

## Acknowledgements

L.D. thanks NSFC (21602099). C.Y.H. thanks R. H. Grubbs (Shenzhen Nobel Prize Scientists Laboratory Project C17783101), Guangdong Provincial Key Laboratory of Catalysis (2020B121201002), SZ research fund (JCYJ 20170817105041557) and SUSTech (Y01501808, Y01506014). We thank Xuran Yao for initial attempts in cycloheptadiene case.

## Author contributions

L.D. and Y.C. developed the chiral catalyst for this methodology, analyzed the results, synthesized and characterized the products obtained. L.D., Y.C., and C.Y.H. participated the discussion.

## Competing interests

The authors declare no competing interests.
