## [Peer Review File · Nature Communications]

Reviewers' comments:

Reviewer #1 (Remarks to the Author):

The author described an enantioselective protocol of hydroalkenylation of cyclic 1,3-dienes and olefin promoted by a chiral NHC–Ni complex. Versatile products 1,4-dienes were produced as an important class of chiral building blocks. The reactions have broad scope with high efficiency, regio- and enantioselectivity. The catalyst is very effective to control all the selectivities. I think this manuscript is suitable for publication on Nature Communications. The following issues should be addressed.

(1) Besides five- and six-membered cyclic 1,3-dienes, can 1,3-dienes in larger ring, such as 7-, 8-, etc. participate in this reaction?

(2) All of the other olefin partner shown contain an allylic heteroatom. Is this necessary for reactivity and selectivity?

(3) The substituents of the alkenes are all electron-donating. Did 1,3-dienes bearing electron-withdrawing groups work in this reaction?

Reviewer #2 (Remarks to the Author):

Ho et al. describe an enantioselective NHC/Ni-catalyzed cross-hydroalkenylation of cyclic dienes and allylic ether as well as allylic amine derivatives. This method provides efficient access to a series of synthetically useful skipped 1,4-diene bearing a chiral allylic center and endocyclic olefin. The use of a C₂-symmetric chiral NHC with less substitution on N-aryls is found as the key for an effective enantioinduction and the control of product distribution. The yields and functional group tolerance are good, the substrate scope is considerably broad, and the levels of regio- and enantiocontrol are generally high. Additionally, they provide a detailed discussion on the substitution effect on the outcomes and the hypothetical models on asymmetric hydronickelation on diene and regioselective insertion of allylic ethers. Although a racemic version of the cross-hydroalkenylation of two similar substrates is known (previously reported by the same authors), the current enantioselective transformation represents an interesting and valuable breakthrough of this chemistry. I think this report is suitable for publication in Nature Communication after the following issues have been carefully addressed.

1. The "alpha-olefin" in the title is not suitable and too broad. alpha-olefins generally mean unactivated simple alkenes. But in this paper, only allylic ether and amine derivatives are included. The use of a single homoallylic ether substrate resulted in low regioselectivity. The effect of heteroatom on alkenes is particularly important to the reaction outcomes. Thus, the description of "alpha-olefin" in the title and the manuscript should be changed.

2. Why 20 mol% of 1-octene is added in the reaction conditions in Table? An explanation is encouraged.

3. I suggest the authors a catalytic cycle provide for better understanding, at least should be included in SI.

4. The enantiomeric excesses of products in Scheme 4 should also be included in the manuscript, not only in SI.

5. The manuscript is generally well written. But many abbreviations that make it inconvenient to read and understand, for example, "chiral P and polar H-X", "M-boryl", and lots of "1", "2" in the main text.

Dear Referees,

Thank you very much for your kind comments, interest and support for the publication of this work (NCOMMS-20-04658A).

We have modified the format of our manuscript as the editorial request (manuscript checklist, e.g. headings/subheadings added, Tables are placed at the back, use of present tense in abstract, reference style, author contributions) and here below is our point-by-point response to the referees' comments:

Reviewer #1 (Remarks to the Author):

Thanks for rating this manuscript is suitable for publication on Nature Communications. We have made the revision according to the advices:

(1) Besides five- and six-membered cyclic 1,3-dienes, can 1,3-dienes in larger ring, such as 7-, 8-, etc. participate in this reaction?

Yes, as shown in Table 2 Set 8, 2-substituted cycloheptadiene is a compatible substrate and 3' was obtained in good yield.

(2) All of the other olefin partner shown contain an allylic heteroatom. Is this necessary for reactivity and selectivity?

Thanks for the question. The heteroatom function was described by means of citation (ref. 26 as shown before), and was illustrated as shown in Table 2 Set 4 as well as Set 6, and in Scheme 2E.

Its main function is related to lowering terminal olefin undesired consumption (ref 26). For its effect on ee, some ee improvement was observed when an optimal substituent was employed (Table 2 Set 4). Isomeric structures of the allyl and homoallyl ether shows comparable ee and yield (Set 6), the change in regioselectivity was discussed in Scheme 2E and relevant text.

Here we emphasized the importance of that by citation and use the description "hetero-substituted terminal olefin" in place of a-olefin that we used earlier.

(3) The substituents of the alkenes are all electron-donating. Did 1,3-dienes bearing electron-withdrawing groups work in this reaction?

1,3-diene bearing electronegative halide as substituent works well as shown in Set 1 with $R^2 = Br$. On the other hand, due to the complications related to in situ enolate/enol formation, carbonyl was found as an incompatible electron-withdrawing group for this work. We added the following: "At this stage, cyclic 1,3-diene substituted with a ketone didn't work well." in the Table 2 result discussion paragraph.

Reviewer #2 (Remarks to the Author):

Thanks for rating the current enantioselective transformation represents an interesting and valuable breakthrough of this chemistry, and suitable for publication in Nature Communication. We have made the revision according to the advices as follow:

1. The "alpha-olefin" in the title is not suitable and too broad. alpha-olefins generally mean unactivated simple alkenes. But in this paper, only allylic ether and amine derivatives are included. The use of a single homoallylic ether substrate resulted in low regioselectivity. The effect of heteroatom on alkenes is particularly important to the reaction outcomes. Thus, the description of "alpha-olefin" in the title and the manuscript should be changed.

Thanks for the advice. We now narrowed it down and emphasized the importance of the heteroatom by using "hetero-substituted terminal olefin" in place of a-olefin that we used earlier.

2. Why 20 mol% of 1-octene is added in the reaction conditions in Table? An explanation is encouraged.

That is used for the generation of NiH from Ni(allyl), which was employed in ref 26-30 as well. Here we emphasized that by adding "(for generation of NiH or equiv.),²⁶⁻³⁰" in the Method section, under the standard procedure of both maintext and SI.

3. I suggest the authors a catalytic cycle provide for better understanding, at least should be included in SI.

A proposed catalytic cycle at this stage was added in the SI as advised. The steps for forming major products are the same as the description in the Discussion section as well as Scheme 2A, and the 1,3-allyl shift as observed in Scheme 3 was emphasized in the catalytic cycle.

4. The enantiomeric excesses of products in Scheme 4 should also be included in the manuscript, not only in SI.

Sorry for the careless mistake. We added it back accordingly.

5. The manuscript is generally well written. But many abbreviations that make it inconvenient to read and understand, for example, "chiral P and polar H-X", "M-boryl", and lots of "1", "2" in the main text.

Thanks for the advice, we now use the full name.

e.g. Phosphorous instead of P, Bpin instead of boryl, cyclic 1,3-diene 1 instead of just 1.

We sincerely thank all of the referees again for the above constructive advice. We hope the revision is sufficient. An annotated manuscript is provided for your kind consideration. No changes in the

SI, except format changes as requested by the editorial office and the added catalytic cycle. Please feel free to contact me if clarification is needed.

Sincerely,

Chun-Yu HO

Shenzhen Grubbs Institute, Department of Chemistry, Southern University of Science and Technology, China.

REVIEWERS' COMMENTS:

Reviewer #1 (Remarks to the Author):

The manuscript should be suitable for publication.

Reviewer #2 (Remarks to the Author):

I suggest publication of this work in Nature Communication.

Some typo and grammar mistakes should be corrected before the publication. For examples, "at here" should be "here", "which have prompted further study in near future" should be "which have prompted a further study in the near future", "work up" should be "workup", "This work is delicated to..." should be "This work is dedicated to...".

Here below are our point-by-point response to reviewers comment:

Reviewer #1 (Remarks to the Author):

The manuscript should be suitable for publication.

Reviewer #2 (Remarks to the Author):

I suggest publication of this work in Nature Communication.

Some typo and grammar mistakes should be corrected before the publication.

For examples, "at here" should be "here", "which have prompted further study in near future" should be "which have prompted a further study in the near future", "work up" should be "workup", "This work is delicated to..." should be "This work is dedicated to...".

Thanks, and those mistakes are now fixed.

The dedication sentence is removed as directed by the Editorial request.

With best wishes,

Dr. Chun-Yu HO

Associate Professor, Email: jasonhcy@sustech.edu.cn

Shenzhen Grubbs Institute, Department of Chemistry, Southern University of Science and Technology.